# Biosorption Characteristic and Cytoprotective Effect of Pb^2+^, Cu^2+^ and Cd^2+^ by a Novel Polysaccharide from *Zingiber strioatum*

**DOI:** 10.3390/molecules27228036

**Published:** 2022-11-19

**Authors:** Wei Jiang, Ying Hu, Zhenyuan Zhu

**Affiliations:** 1Laboratory of Food Nutrition and Safety, Ministry of Education, Tianjin Key Laboratory of Food Nutrition and Safety, Tianjin University of Science and Technology, Tianjin 300457, China; 2College of Food Science and Engineering, Tianjin University of Science and Technology, Tianjin 300457, China; 3Department of Health Management, Zunyi Medical and Pharmaceutical College, Zunyi 563006, China; 4College of Public Health, Zunyi Medical University, Zunyi 563006, China

**Keywords:** *Zingiber strioatum*, polysaccharide, heavy metal, biosorption

## Abstract

The pollution of heavy metal ions can cause damage to the human body through food, so developing a new biocompatible material that can remove the damage of heavy metal ions has a good application prospect. In this study, we obtained a new homogeneous polysaccharide composed of seven monosaccharides from *Zingiber strioatum* by using the method of separation and purification of polysaccharide. The results of adsorption behavior showed that the concentration, temperature and pH value could affect the adsorption effect of *Zingiber strioatum* polysaccharide (ZSP). Through model fitting of the data of adsorption time and metal concentration, the pseudo second-order kinetic model can well describe the kinetics of the adsorption process, and the adsorption isotherm data fit well with the Langmuir model. In the preliminary research results of adsorption mechanism, SEM showed the appearance of ZSP as flake and porous surface; EDX analysis confirmed the metal adsorption capacity of ZSP. Meanwhile, XPS and FT-IR further clarified the adsorption based on functional groups composed of C and O. The cells preprotected by ZSP can resist heavy metal ions. The above results show that ZSP can be used as a new macromolecule to bind heavy metal ions, which can broaden the research scope of polysaccharides in contaminated food systems.

## 1. Introduction

The industrialization process has led to the occurrence of heavy metal pollution, which has a negative impact on human health [1,2]. As is known to all, heavy metals are non-biodegradable and can be accumulated into organisms through food chains. The treatment of heavy metal poisoning mainly lies in removing the accumulated heavy metals in the human body and reducing the toxic effect of heavy metals on organisms. The classical treatment method mainly uses chelating agent to adsorb heavy metal ions. Polymer polysaccharides have special chemical bonds and functional groups, so they can remove heavy metals. Many studies have also confirmed that polysaccharides can adsorb heavy metals [3]. Therefore, using polysaccharide to promote the excretion of these toxins from the human body is a very promising method.

The adsorption of heavy metals by polysaccharides has been studied by many scholars. However, the adsorption of heavy metals by polysaccharides reported in most of the literature mainly focuses on fungal polysaccharides [4,5,6,7], bacterial extracellular polysaccharides [8,9], chitosan [10], etc. In present studies, the adsorption mechanism of polysaccharides for heavy metals is attributed to the combination of chemical groups (such as hydroxyl, carbonyl, sulfhydryl, etc.) and special spatial structure of polysaccharide with metal ions to form complexes. On the other hand, plant polysaccharides have become a research hotspot because of their biological activities, such as being an antioxidant, hypoglycemic, anti-inflammatory and regulating gut flora [11,12,13]. At present, there are few relevant literature reports on heavy metal adsorption capacity of polysaccharides from plants and the adsorption mechanism is not clear.

In this paper, we propose a new biosorbent, which is derived from natural plant resources and can be used for the biosorption of heavy metals in aqueous solutions. The adsorption behavior of ZSP was investigated mainly through the changes of pH, time, temperature and concentration during the adsorption process. SEM, EDX, XPS and FT-IR were used to reveal the adsorption mechanism of ZSP. The purpose of the above research is to provide a theoretical basis for ZSP to develop an edible biomaterial to remove heavy metal ions from drinking water or food.

## 2. Results and Discussion

### 2.1. Purification and Composition of ZSP

The separated ZSP is a light white, tasteless flocculent and easy to be dissolved in water (Appendix A). The UV scanning spectral analysis of ZSP was shown in Appendix A. There was no absorption peak at 260 or 280 nm, indicating an absence of nucleic acid and protein [14]. The HPLC chromatogram of ZSP molecular weight distribution is shown in Figure 1. The total sugar content of ZSP were approximately 90.6%. As shown in Figure 2, the monosaccharide composition of ZSP was chiefly constructed of galactose (36.8%), mannose (22.8 %) and glucose (20.7%) in association with a small number of xylose (9.8%), arabinose (4.3%), glucuronic acid (2.9%) and galacturonic acid (2.7%) units. In addition, we have reported the molecular weight size (1.57 × 10^6^ Da) and structural composition of ZSP (seven linkages: Manp-(1→,→2)-Xylp-(1→,→6)-Glcp-(1→,→4)-Arap-(1→,→3,4,6)-Glcp-(1→,→2)-Glcp-(1→and Xylp-(1→respectively, in which→4)-Galp/GalpA-(1→might be the main linkage) in another paper which has not been published. It has been reported that the adsorption of heavy metals by polysaccharides may be related to the structure of polysaccharides (e.g., glucan) [6,15]. However, uronic acid can provide carboxyl group as a negatively charged group [16], which has been proved to be able to combine with positively charged metal ions. Therefore, the special structure of ZSP provides conditions for its adsorption of metal ions.

### 2.2. Adsorption Behaviors

#### 2.2.1. Effect of Initial ZSP Concentration

The influence of varying ZSP concentrations were presented in Figure 3A. Results indicated an increase in the adsorption capacity of Pb^2+^, Cu^2+^ and Cd^2+^ within concentration ranges (0.2–0.8 g/L) and the maximum adsorption values are 29.61 mg/g, 20.05 mg/g and 23.13 mg/g at concentration of approximately 0.8 g/L. It could be attributed to more available binding sites for metal ions biosorption presented by the increased ZSP concentration [17]. The adsorption capacity tends to balance between 0.8–1.2 g/L. It may be that the binding sites of ZSP overlap each other at high concentration to form a barrier, which hinders the binding with metal ions [15]. The adsorption capacity of ZSP for the three heavy metals in descending order is Pb^2+^ > Cd^2+^ > Cu^2+^. In addition, the radii of the three metal ions are Pb^2+^ (119 pm) > Cd^2+^ (97 pm) > Cu^2+^ (73 pm). With the same charged ions, the smaller the ionic radius, the larger the radius of hydrated ions. Therefore, the settling capacity in the system is reduced.

#### 2.2.2. Effect of Temperature on Pb^2+^, Cu^2+^ and Cd^2+^ Adsorption

Considering that temperature fluctuations affect the metal adsorption process [18], the effect of temperature on Pb^2+^, Cu^2+^ and Cd^2+^ adsorption capacity of ZSP were explored in a range of 15 °C to 55 °C (Figure 3B). With the increase in temperature, the intermolecular motion accelerates and the contact between ZSP and metal ion adsorption sites is accelerated [19]. The temperature of the medium affects the adsorption capacity of biomass for heavy metal ions from aqueous media. The adsorption capacity was improved in the temperature range of 15–35 °C, which indicates that the adsorption process is heat absorbing in nature [20]. However, if the temperature is too high (over 35 °C), the adsorption capacity does not increase significantly. Hence, we infer that the continuous increase in temperature may lead to the breaking of some functional groups^,^ bonds or the reduction in active adsorption sites, resulting in the fact that the adsorption capacity does not enhance significantly with the increase in temperature [21,22].

#### 2.2.3. Effect of pH on Pb^2+^, Cu^2+^ and Cd^2+^ Adsorption

The pH value is an important factor controlling the adsorption capacity [23]. As shown in Figure 3C, the extent of removal of Pb^2+^, Cu^2+^ and Cd^2+^ were investigated by varying the pH in the range of 1.0 to 6.0. At a lower pH (pH = 1.0–4.0), the biosorption of metal ions were very little, due to a large number of H+ ions in the solution occupy the binding sites of ZSP in advance and form a competitive relationship with metal ions [24]. On the other hand, we can see that there was almost no variation in adsorption capacity of lead, copper and cadmium with increasing the pH of the metal solution during the biosorption process up to pH = 4.0–6.0 and the maximum adsorption capacity was obtained at pH = 4.0. The increase in Pb^2+^, Cu^2+^ and Cd^2+^ biosorption when the pH increases, may be due to the decrease in protonation of functional groups of ZSP, the negative charge density on the surface increases, so the interaction between metal cations and the negative charges of ZSP functional groups, the adsorption capacity gradually increases [25]. It has been reported in the relevant literature that with increasing pH (at pH close to 7), many unstable hydrolysates and polymeric substances (e.g., Pb(OH)^+^, Pb(OH)_2_, [Pb_3_(OH)_5_]^+^, [Pb_4_(OH)_4_]^4+^, [Cu_2_(OH)_2_]^2+^, [Cu_3_(OH)_4_]^2+^, Cu(OH)^+^, [Cu_2_(OH)]^3+^, Cu(OH)_2_, Cd(OH)^+^, etc.) may exist for the three metal ions and the formation of these substances may be associated with enhanced binding to ZSP. At a pH greater than 7, the metal cations may precipitate due to the formation of hydroxyl species [26,27].

#### 2.2.4. Adsorption Kinetics Study

Based on the optimal adsorption conditions (biomass addition is 0.8 g/L; reaction temperature is 25 °C; reaction pH is 4.0), we investigated the influence of contact time on the adsorption behavior and the results are shown in Figure 3D. It was observed that the maximum removal Pb^2+^, Cu^2+^ and Cd^2+^ ions of ZSP occurred in about 40 min. After this period of time, the number of bound metal ions showed a relatively balanced trend in the reaction process. The kinetics results of Pb^2+^, Cu^2+^ and Cd^2+^ are shown in Table 1 and Appendix A. On the basis of the higher R^2^ value, the pseudo second-order model gave a better fit to the biosorption data than did the pseudo first-order model. This also shows that chemical adsorption is the rate determining step of the adsorption process [26].

#### 2.2.5. Biosorption Isotherm Study

Effect of initial metal solutions concentration on adsorption behavior is shown in Figure 3E. Optimum conditions of the biomass addition 0.8 g/L, temperature 25 °C, pH = 4.0 and time 45 min were used for isothermal adsorption study. With the increase in the concentration of Pb^2+^, Cu^2+^ and Cd^2+^ metal solutions, the adsorption capacity of ZSP to the three metals is gradually enhanced. Meanwhile, it was found that the removal rates of metal ions by ZSP decreased or tend to balance with an increase in metal ion concentrations after 75 mg/L, 100 mg/L and 100 mg/L for Pb^2+^, Cu^2+^ and Cd^2^, respectively. The reason for this phenomenon is that the surface of ZSP has sufficient functional groups and can adsorb metal ion with low concentration. With the continuous increase in initial metal ion concentration, adsorption was difficult since these adsorption sites were already occupied by previously sorbed metal ions. The results of Table 2 and Appendix A show that the Langmuir isotherm model shows better correlation than the Freundlich isotherm model, which indicates the adsorption process was a monolayer [28]. The maximum biosorption capacity on Pb^2+^, Cu^2+^ and Cd^2+^ was 67.73 mg/g, 59.34 mg/g and 65.05 mg/g, respectively.

### 2.3. Preliminary Study on Adsorption Mechanism

#### 2.3.1. The Surface Structure and EDX Analysis of ZSP

It can be seen from SEM images (Figure 4), ZSP is stacked together in flakes, distributed in a scattered manner and there are holes in the fragments. Such lamellar structures may provide favorable conditions for ion exchange, complexation and coordination in the process of metal adsorption.

EDX spectra is often used to verify the combination of metal ions and biomass [16]. EDX spectra indicated the main elements for ZSP are C, O and N (Figure 5). After three metal ions adsorption, new peaks emerged, indicating successful binding of Pb^2+^, Cu^2+^, Cd^2+^ with ZSP.

#### 2.3.2. X-ray Photoelectron Spectroscopy (XPS) Analysis of ZSP

The XPS spectra of ZSP are shown in Figure 6, which represent the whole XPS measurement range. The results show that there is no Pb^2+^, Cu^2+^, Cd^2+^ on the surface before adsorption of ZSP, because the binding energy of these three metal ions are not detected. After heavy metal ions are absorbed by ZSP, the binding energies of 162 ± 1.0 eV (Pb4f line), 935 ± 1.0 eV (Cu2p line) and 411 ± 1.0 eV (Cd3d line) are detected by XPS, which also indicated that the surface of ZSP has been loaded with Pb^2+^, Cu^2+^ and Cd^2+^. It is reported in the literature that the change of C1 peak is due to the change of peak area or binding energy after cation is absorbed [29]. Meanwhile, after the absorption of the three metal ions, the peaks of C1s (275.16 eV) and O1s (527.15 eV) show varying degrees of weakening, indicating that the functional groups related to C and O (O-H, C-OH and C=O) can effectively bind to Pb^2+^, Cu^2+^ and Cd^2+^, which is consistent with the results reported in the literature [30,31]. This result can be further verified by infrared spectrum scanning.

#### 2.3.3. FT-IR Spectroscopy Analysis of ZSP

In Figure 7, the difference before and after metal adsorption by ZSP is analyzed by FT-IR spectrum. The peaks assignments of the ZSP were as follows: the ZSP showed a broad band at about 3423.16 cm^−1^, assigned to stretching vibration modes of O-H groups [32,33]. The weak peak toward 2923.83 cm^−1^ were related to the stretch vibration of C-H bond [34] and the signals at 1850–1600 cm^−1^ were due to the stretch vibration of C=O bond [35]. The peak at about 1030 cm^−1^ speculated to the C-OH stretching and the signals at about 900–1150 cm^−1^ for C-O-C bond [36]. The band observed at 1036 cm^−1^ changed to 1030 cm^−1^ may indicate the presence of O-H groups (cellulosic compounds) or phosphate group in the biomass [37].

After loading Pb^2+^, Cu^2+^ and Cd^2+^, the peak distribution wave number of ZSP changed significantly. Generally speaking, the biochemical reaction between ZSP and metal ions changes the chemical environment of functional groups, resulting in corresponding band shift (Table 3). The results of infrared spectra show that O-H, C=O, C-OH, C-O-C and phosphate perform an important role in the binding of ZPS with Pb^2+^, Cu^2+^ and Cd^2+^. Bhunia et al. believed that the binding ability of polysaccharides depended on the expression of negatively charged groups, such as hydroxyl, carbonyl, alcohol, amine, phosphoryl, sulfhydryl and carboxyl on the surface of polysaccharides [16]. Obviously, our FTIR analysis also confirmed this view.

### 2.4. Protective Effects of ZSP on Cells Treated with Pb^2+^, Cu^2+^ and Cd^2+^

It can be seen from Figure 8 that the survival rate of HepG2 cells in ZSP preprotected group increased significantly. However, in the treatment group with heavy metal ions added without protection, the cell survival rate was significantly reduced. Li et al. used oyster ferritin (GF1) to pre-protect HFF-1 cells and MC3T3-E1 cells and found that GF1 had a strong protective effect against heavy metal invasion due to its strong ability to enrich heavy metal ions [38]. It is no coincidence that ZSP polysaccharide has been shown to have an adsorption ability for heavy metals and has no toxic effect on cells. Therefore, when heavy metal solutions were added to the cell culture medium, the pre-protective effect was exerted by the adsorption of heavy metals by the pre-added ZSP, which reduced the concentration of heavy metals in the medium or produced substances that could not cause cellular toxicity. This fact may imply a reduced risk of cellular damage. Meanwhile, it also proved that ZSP has a relatively strong protective effect against the invasion of heavy metal ions.

## 3. Materials and Methods

### 3.1. Zingiber strioatum and Chemical Reagents

Fresh *Zingiber strioatum* were harvested in Qixingguan County, Bijie City, Guizhou Province, China, in September 2020. The harvested *Zingiber strioatum* were dried (65 °C), crushed into powder (100 mesh), sealed in plastic bags and stored in a cool and dry place before extraction. All other chemicals and reagents were purchased locally and were of analytical grade.

### 3.2. Biosorbent Preparation

The dry *Zingiber strioatum* powder (30 g) was mixed with distilled water (1:20, *w*/*v*), extracted three times at 80 °C for 2 h. The extract solution was filtered and concentrated (evaporated to 1/4 of the original volume under reduced pressure to remove excess water) and precipitated with ethanol until reaching a final concentration of 80% (*v*/*v*) under vigorous stirring. After overnight storage at 4 °C, the precipitate was obtained by centrifugation (4000 r/min, 15 min). Subsequently, the sevage reagent, which was composed of chloroform and n-butanol (4:1, *v*/*v*), was applied to remove the protein from the washed precipitate. In addition, the ultraviolet absorption spectrum (190 nm to 400 nm) was used to detect whether the protein in the polysaccharide was removed completely. The sevage reagent was removed by vacuum concentration, and crude polysaccharide was obtained after freeze-drying. Finally, the polysaccharide was purified by DEAE Cellouse-52 (Particle size: 50 µM; ligand density: 40 µmol/mL; adsorption capacity: 110 mg HSA/mL) and Sephadex G-150 column (Globulin separation range: 5000~3 × 10^5^). Through such separation and purification processes, a pure ZSP was obtained. With D-glucose as the standard, the total carbohydrate content was determined by phenol sulfuric acid method. The absorbance was detected at 490 nm and the glucose standard curve was drawn. The monosaccharide composition of ZSP was evaluated using GC-MS with some modifications [39].

### 3.3. Metal Solution Preparation

Solutions of Pb^2+^, Cu^2+^ and Cd^2+^ ions were prepared by dissolving predefined amounts of CuSO_4_, Pb(NO_3_)_2_ and Cd(NO_3_)_2_, respectively, in distilled water, to achieve ion concentrations of 1 g/L in each flask. The metal ions with different initial concentrations were prepared by diluting the stock solution. The pH of the solutions was adjusted using HCl (0.1 mol/L) or NaOH (0.1 mol/L) to achieve the desired values.

### 3.4. Adsorption Behaviors

In adsorption experiment, ZSP was added to 50 mL centrifuge tube with metal solution and then placed on an orbital shaker (Tianjin Weihua Experimental Instrument Factory, Tianjin, China) for continuous shaking at 300 r/min. Each of the sets were prepared in triplicates. For preparation of the control, only ZSP was added to deionized water for sample 1 and in sample 2 only respective metal ion solutions were used. Supernatant was obtained after centrifugation (4000 r/min, 10 min) of sample solution. The supernatant was passed through a 0.22 µM filter membrane (Tianjin Jinteng Experimental Equipment Co., Ltd., Tianjin, China) and determined by atomic absorption spectrophotometer (Beijing Puxa, Beijing, China). The adsorption capacity of the metal ion adsorbed (*q_e_*) was calculated as follows:(1)$$q_{e}=(C_{0}-C_{t})\times V/m$$
where, *C*_0_ and *C_t_* are the initial and final concentrations of the metal ion in solution (mg/L), *V* is the solution volume (L) and *m* is the mass of the sorbent (g).

#### 3.4.1. Effect of ZSP Concentration

The effect of dose of ZSP on metal ion biosorption was studied in order to obtain the optimal concentration of adsorbent. The respective metal ion solutions (10 mL each; 50 mg/L) were mixed with ZSP polysaccharide solutions of different concentrations (0.2–1.2 g/L) at room temperature.

#### 3.4.2. Effect of Temperature

ZSP (8 mg each) were added to metal solution (10 mL each; 50 mg/L) and then placed on an orbital shaker for continuous shaking at temperature range (15–55 °C) for 90 min. In this experiment, the effect of temperature on the adsorption performance of ZSP was investigated to obtain the optimal reaction temperature.

#### 3.4.3. Effect of pH

The pH of metal ion solutions was 1.0–6.0. ZSP (8 mg each) and were added to the solutions (10 mL each; 50 mg/L) at 25 °C for 90 min. Here, the change in ZSP adsorption capacity due to the change in pH was examined in order to obtain the optimal reaction conditions.

#### 3.4.4. Adsorption Kinetics

Based on the determination of the optimal biomass addition amount, reaction temperature and reaction pH, the effect of reaction time (5–120 min) on adsorption capacity was investigated. The experimental data were used for adsorption kinetics equation fitting analysis [40]. The pseudo-first and pseudo-second order kinetic equation are separately defined as:(2)$$q_{t}=q_{e}(1-e^{-k_{1}t})$$
(3)$$q_{t}=tq_{e}{}^{2}K_{2}/(tK_{2}q_{e}+1)$$
where, *q_t_* is the adsorption amount of metal ions at time *t* (mg/g), *q_e_* is the saturated adsorption amount of metal ions (mg/g), *K*_1_ and *K*_2_ describe the rate constant.

#### 3.4.5. Isothermal Adsorption

The effect metal ion concentration (15–200 mg/L) has on adsorption capacity of ZSP was studied. Optimum conditions, as described in Section 2.4, of the biomass addition, temperature, pH and time were used for isothermal adsorption study. The experimental data were used for the fitting analysis of isotherm models [41].

Langmuir isotherm adsorption equation is defined as:(4)$$q_{e}=q_{m}K_{L}C_{e}/(K_{L}Ce+1)$$

Freundlich isotherm adsorption equation is defined as:(5)$$q_{e}=K_{F}{C_{e}}^{1/n}$$
where, *C_e_* is the maximum concentration of adsorbed metal ions (mg/L), *q_m_* is the maximum adsorption under ideal conditions (mg/g), *K_L_* is the Langmuir constant, 1/n is the exponent of adsorption ability and *K_F_* describes the Freundlich isotherm constant.

### 3.5. Preliminary Study on Adsorption Mechanism

#### 3.5.1. Surface Characteristics of ZSP and Energy Dispersive X-ray Detector (EDX) Analysis

Before the experiment, all samples are temporarily stored in a glass dryer after vacuum freeze-drying for standby. ZSP samples before and after adsorption were fixed with conductive adhesive. Scanning electron microscopy (SEM) images of the lyophilized ZSP were recorded using a SU1510 electron microscope (Hitachi, Tokyo, Japan). Prior to measurements, a thin gold film was coated on the surface of the specimens to enhance conductivity. Working conditions: the acceleration voltage is 15 kV, the magnification is 500 times respectively, and a representative field of view is selected for photographing and recording [42]. The surface morphology of adsorbed metal and elemental composition of the samples were studied by SEM using a microscope equipped with EDX analysis.

#### 3.5.2. X-ray Photoelectron Spectroscopy (XPS) Analysis

The XPS was performed using an optoelectronic ESCA spectrometer (Shimadzu Kratos axis ultra, UK) equipped with monochrome Al (Kα). The XPS instrument of X-ray excitation source operates at 300 W.

#### 3.5.3. Infrared Spectroscopy (FT-IR) Analysis

Accurately weighed 1.00 mg of each sample was ground with dry KBr powder (100 mg) in a grinder until a uniform powder was obtained. Powder of respective sample was placed in the mold and pressed into transparent sheets with a tablet press. The structure of the samples was investigated by FT-IR employing a Vector 22 Fourier transform infrared spectrometer (Bruker, Frankfurt, Germany) operated in the region of 400–4000 1/cm. The infrared spectra were collected at a resolution of 2 1/cm with 16 scans.

### 3.6. Cytoprotective Effect of ZSP

Once heavy metals enter the human body, they are difficult to eliminate and are mainly metabolized by the liver. Excessive accumulation of heavy metals will damage the liver. Therefore, HepG2 cell was selected and preprotected with ZSP to evaluate the ability of polysaccharide to bind heavy metal ions in vitro. The method of Li et al. was referenced and slightly modified [38]. The cells were cultured in DMEM medium containing 10% fetal bovine serum, penicillin and streptomycin at 37 °C and 5% CO_2_. When the cells grew to about 80% of the coverage rate, they were cultured with trypsin enzyme digesting until the cells grew to logarithmic phase and then the cells were transferred to 96-well plates (cell suspension is 100 µL/hole). The ZSP solutions (10 µL/hole) were added to the cells for preprotection and subsequent experiments were conducted after the cells grew for 3 h. The metal ion solutions (10 µL each; 20 mM) were added to the sample holes of cell culture plate respectively and the protective effect of ZSP on cells was observed after 24 h. The cellular activity was determined by CKK-8 assay.

### 3.7. Statistical Analysis

The results were expressed as mean ± SD. Every data point was analyzed (Duncan’s test) using SPSS 26 (SPSS, IBM, Frisco, TX, USA). The difference was considered statistically significant when *p* < 0.05. SPSS was also used for linear fitting of adsorption kinetic model and isothermal adsorption model.

## 4. Conclusions

The food safety problem caused by heavy metal pollution of drinking water and food has seriously affected human life and health. Therefore, finding a safe and effective method to remove heavy metal ions has become a research hotspot. A polysaccharide (ZSP) was isolated and purified from *Zingiber strioatum*. ZSP has the ability to adsorb lead, copper and cadmium metal ions, but the adsorption performance is influenced by conditions, such as adsorbent concentration, pH, contact time, temperature and metal ion concentration. The adsorption of heavy metals by ZSP may depend mainly on the lamellar porous structure and its own functional groups composed of C and O. Adsorption isotherms and kinetic models also predicted that monomolecular layer binding chemical group adsorption was the main mode of ZSP adsorption. In vitro cell experiments also confirmed that ZSP can reduce the risk of cell damage when heavy metals invade. ZSP can be used as a new edible material for the removal of heavy metal ions from drinking water, while its adsorption of heavy metals in vivo and its protective effect on organs will be investigated in the future.

## Figures and Tables

**Figure 1 molecules-27-08036-f001:**
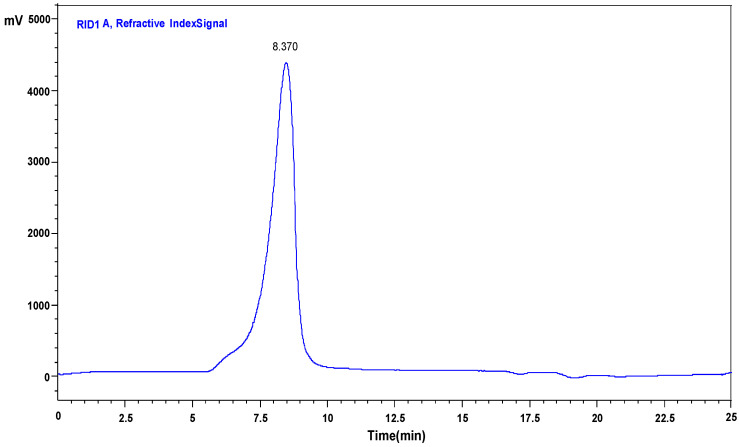
HPLC chromatograms of ZSP about molecular weight distribution.

**Figure 2 molecules-27-08036-f002:**
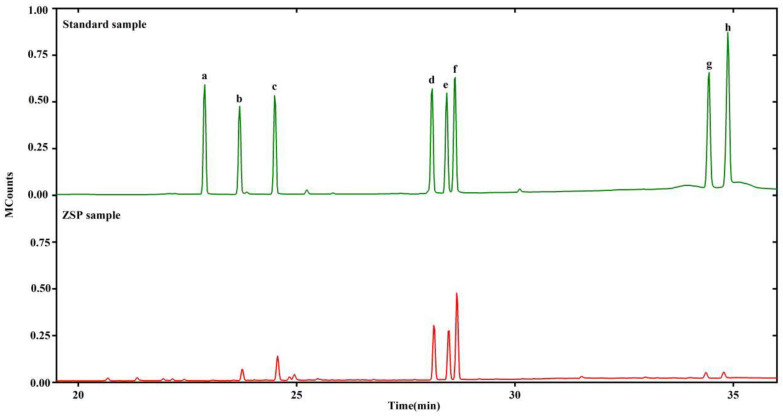
Standard monosaccharides: (**a**) D-rhamnose, (**b**) D-arabinose, (**c**) D-xylose, (**d**) D-mannose, (**e**) glucose, (**f**) D-galactose, (**g**) D-galacturonic acid, (**h**) D-glucuronic acid (Above); GC-MS profile of ZSP peaks (Below).

**Figure 3 molecules-27-08036-f003:**
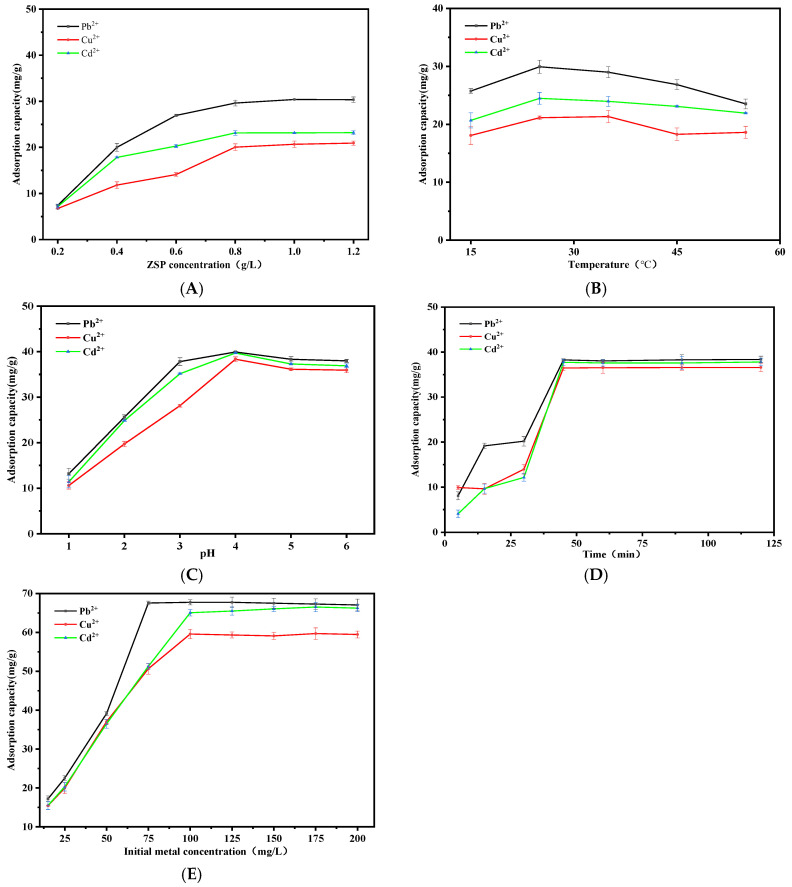
Influence of ZSP concentration (**A**), temperature (**B**), pH (**C**), time (**D**) and initial metal concentration (**E**) on adsorption of heavy metal ions.

**Figure 4 molecules-27-08036-f004:**
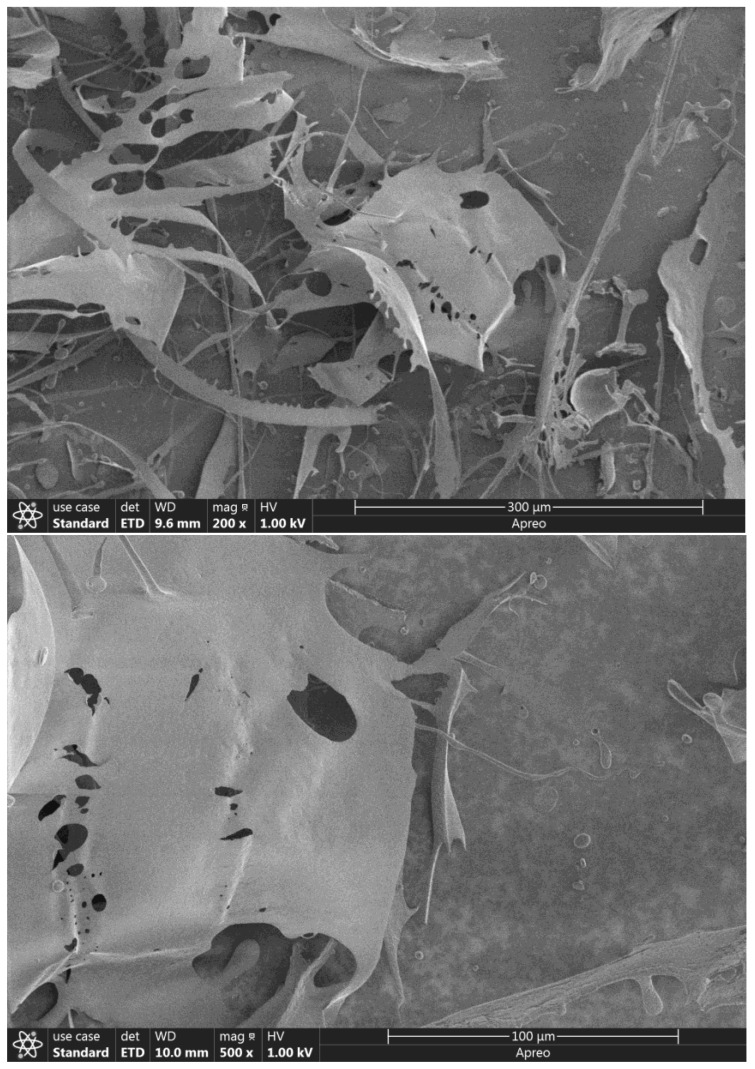
SEM photographs of ZSP(200× and 500×).

**Figure 5 molecules-27-08036-f005:**
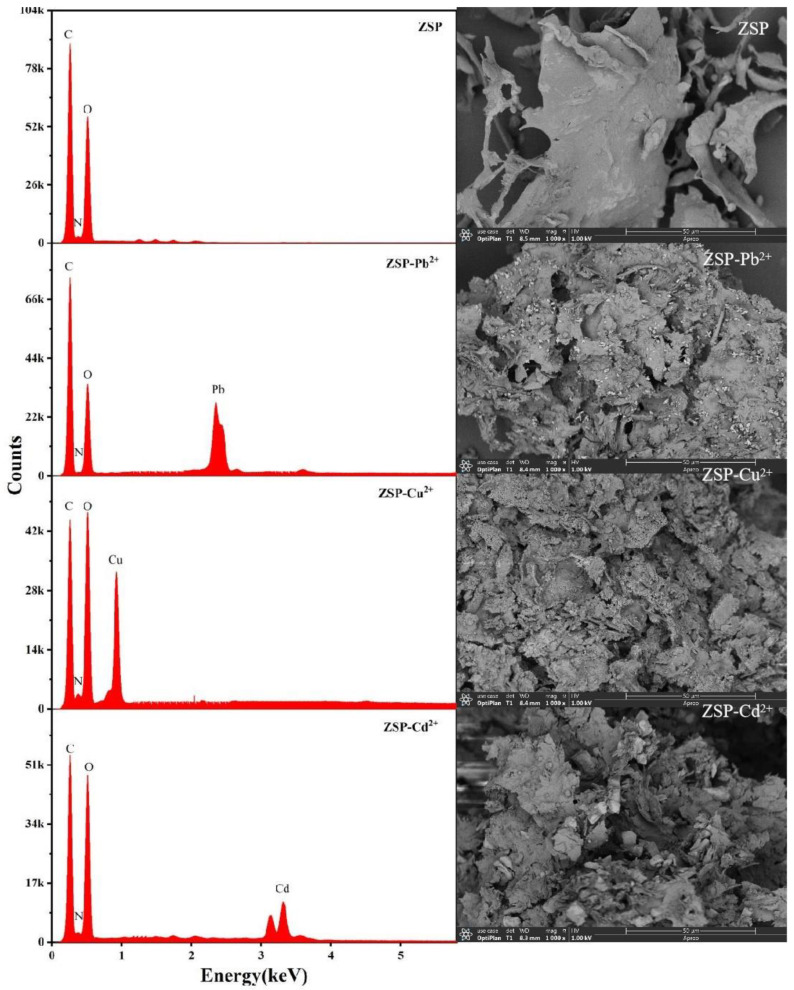
EDX/SEM spectra of before and after the adsorption of Pb^2+^, Cu^2+^ and Cd^2+^ by ZSP.

**Figure 6 molecules-27-08036-f006:**
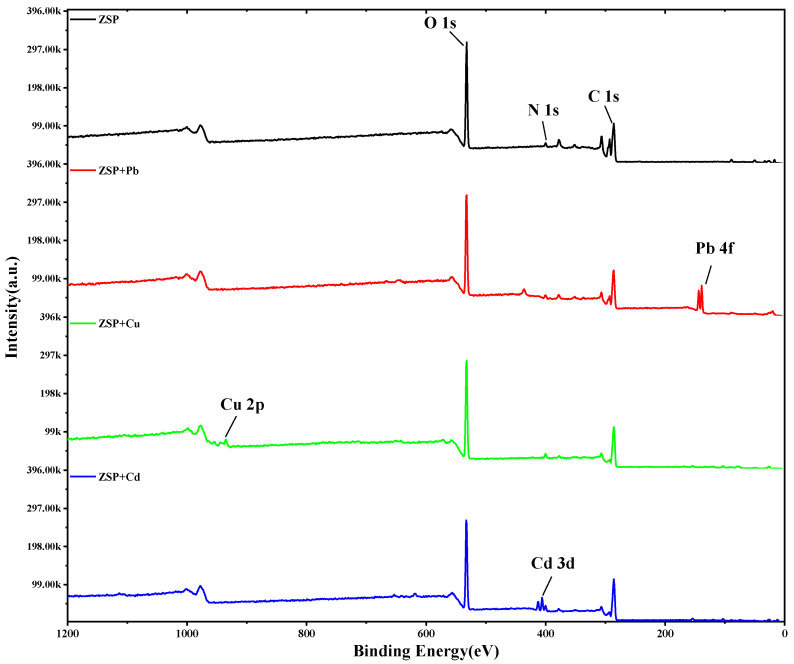
XPS broad scan spectra of ZSP before and after adsorption of Pb^2+^, Cu^2+^ and Cd^2+^.

**Figure 7 molecules-27-08036-f007:**
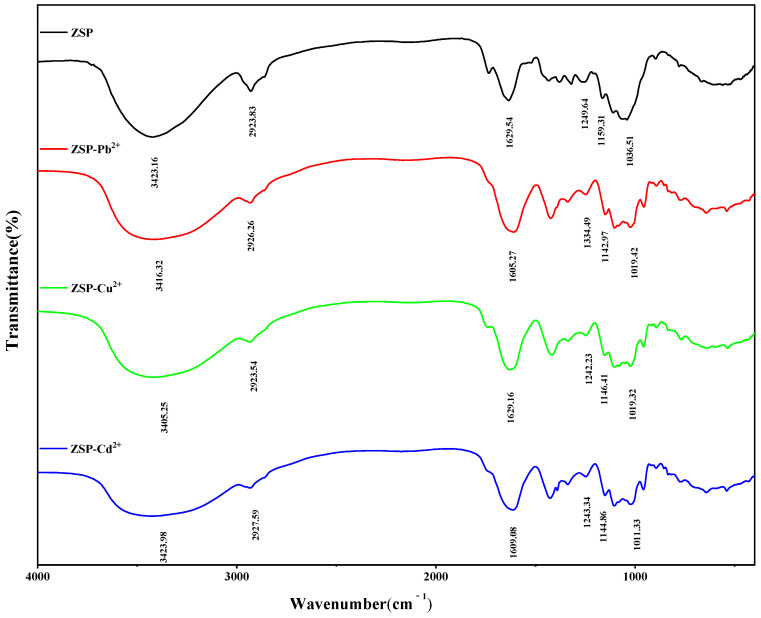
FT-IR spectra of ZSP and ZSP loaded with Pb^2+^, Cu^2+^ and Cd^2+^.

**Figure 8 molecules-27-08036-f008:**
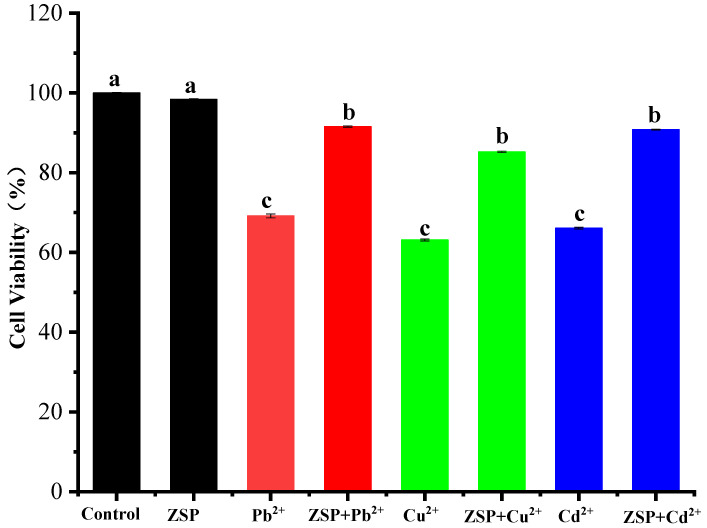
Preprotective effect of ZSP on cells treated with Pb^2+^, Cu^2+^ and Cd^2+^. The cells were preprotected by ZSP polysaccharide for 12 h and treated with heavy metal ions for 12 h. Cell viability was measured by CKK-8 assay.(“a–c” means significant difference between different treatment groups; *p* < 0.05.)

**Table 1 molecules-27-08036-t001:** Kinetic model parameters.

	Pseudo First-Order Model			Pseudo Second-Order Model		
	K_1_	*q_e_*	R^2^	K_2_	*q_e_*	R^2^
ZSP-Pb^2+^	0.006	18.93	0.6758	0.0015	40.32	0.9919
ZSP-Cu^2+^	0.035	10.93	0.6579	0.0024	20.53	0.9925
ZSP-Cd^2+^	0.029	13.79	0.6770	0.0009	33.89	0.9515

**Table 2 molecules-27-08036-t002:** Isotherm model parameters.

	Langmuir Isotherm Model			Freundlich Isotherm Model		
	Q_m_ (mg/g)	K_L_ (L/mg)	R^2^	K_F_ (mg/g)	n	R^2^
ZSP-Pb^2+^	67.73	0.2478	0.9969	17.95	2.879	0.8820
ZSP-Cu^2+^	59.34	0.2780	0.9721	13.78	2.358	0.9128
ZSP-Cd^2+^	65.05	0.2521	0.9653	19.13	2.160	0.9456

**Table 3 molecules-27-08036-t003:** Wave numbers (cm^−1^) of dominant peaks obtained from FT-IR transmission spectra of ZSP + Pb^2+^, Cu^2+^, Cd^2+^.

Sample	Functional Groups					
	O-H	CH_2_	C=O	C-OH	C-O-C	O-H or Phosphate
ZSP	3423.16	2923.83	1629.54	1249.64	1159.31	1036.51
ZSP + Pb^2+^	3416.32	2926.26	1605.27	1243.11	1142.97	1019.42
ZSP + Cu^2+^	3405.25	2923.54	1629.16	1242.23	1146.41	1019.32
ZSP + Cd^2+^	3423.98	2927.59	1609.08	1243.34	1144.86	1011.33

## Data Availability

Not applicable.

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
