# Peer review of "Biosorption Characteristic and Cytoprotective Effect of Pb2+, Cu2+ and Cd2+ by a Novel Polysaccharide from Zingiber strioatum"

_molecules, 2022, doi:10.3390/molecules27228036_

Round 1
Reviewer 1 Report
The manuscript "Biosorption characteristic and cytoprotective effect of Pb2+, Cu2+ and Cd2+ by a novel polysaccharide from Zingiber strioatum" shows the results of an interesting and well-developed work that provides novel information on the potential applications in public health of a polysaccharide of plant origin and easy to get. However, there are some minor aspects that the authors should improve before it is accepted for publication in Molecules.
Comments:
1. Adsorption tests should be better explained in point 2.4 since it is the point that introduces the following procedures. Therefore, the type of flasks used and their volume, the rpm used in agitation, the type of membrane of the filters used to collect the sample and also the description of the AA equipment used to analyze the samples should be detailed. A better explanation of the general procedure will allow an optimal understanding of important aspects, for example, the authors describe the use of a "control" that does not carry heavy metals but in adsorption tests the control is usually the original metal solution for compare their final concentration after completion of the experiment. Please explain better all these aspects.
2. As a suggestion, between points 2.4.1 and 2.4.3 the authors explain the dynamics of the tests to determine the influence of biomass concentration, temperature and pH; it is a sequential protocol that should be improved in the future by designing experiments. At the same time, section 2.4.3 does not specify the concentration of biomass used in the tests, nor the volume of metal solution. In addition, these three sections must specify the ultimate purpose of the experiments described: obtain the optimal operating conditions?
3. In section 2.4.4 the authors describe the procedure to obtain the kinetics, but do not detail the operating conditions. I imagine that it was the optimal conditions obtained in the previous tests, but this aspect must be perfectly explained. I understand that these “optimal conditions” are results and cannot be described in this section, but this important aspect must be clearly specified.
4. The foregoing is applicable to point 2.4.5.
5. Sections 2.5.1 and 2.5.2 should be joined because the EDX sensor is attached to the SEM. In addition, the sample preparation protocol must be included before metallization with Au, that is, fixation and desiccation steps, if any. Any author who wants to reproduce the procedure must be clear about the steps to follow.
6. Section 2.5.3, something to say about sample preparation?
7. Section 2.6, specify the volume of cell suspension and ZPS suspension added to the wells. Also the volume of metallic solutions. Although this can be inferred, please specify how the controls were designed.
8. Section 3.2.3, the operating conditions do not appear (see comment no. 3).
9. In section 3.2.4 it is important not to confuse “optimal biosorption capacity” with “maximum biosorption capacity”, which is a theoretical value produced by the Langmuir model. Please correct the error. In addition to the above, what was said in point 3 is also valid for this section since we do not know what operating conditions were used to obtain the isotherms, apart from the metal concentrations.
10. Sections 3.3.1 and 3.3.2 should be merged into a single section, titled for example “SEM-EDX Analysis”.
11. Section 3.4, the results should be better discussed in the real context of the findings. It is not possible to state that ZSP can inhibit cell death induced by heavy metal ions, in fact ZPS is capable of adsorbing heavy metals thus reducing their concentration in the medium and secondarily this fact may mean a reduction in the risk of cell damage. This comment is also applicable to the Conclusions section. On the other hand, supporting the comments with a bibliography can improve the discussion of section 3.4.
12. Unify the criteria established in the introduction with the final conclusions. In the introduction, the use of ZPS for the treatment of water contaminated by heavy metals destined for the food industry and agriculture is identified as the objective of the work. The objectives and conclusions must be related, building a coherent argument from the experimental findings.
13. Unify the font size in the figure captions.
Author Response
Responses to reviewers, comments
Thank you very much for your affirmation of our work and kindly comments. We have revised the manuscript carefully. The responses and modifications are as follows according to your comments.
All revisions were marked in blue in the revised manuscript and Supplementary Materials
Responds to comments from Reviewer 1
Q1. Adsorption tests should be better explained in point 2.4 since it is the point that introduces the following procedures. Therefore, the type of flasks used and their volume, the rpm used in agitation, the type of membrane of the filters used to collect the sample and also the description of the AA equipment used to analyze the samples should be detailed. A better explanation of the general procedure will allow an optimal understanding of important aspects, for example, the authors describe the use of a "control" that does not carry heavy metals but in adsorption tests the control is usually the original metal solution for compare their final concentration after completion of the experiment. Please explain better all these aspects.
Response: Thank you for pointing out this. We have re described the type and volume of the container used, the instrument used for stirring and the rotating speed used, the type of membrane of the filter used to collect samples, and the description of the atomic absorption spectrometer used to analyze samples, which are marked in blue in the manuscript. Moreover, very sorry for our insufficient statement. The phrase “The control was one sample without heavy metal ions.” was changed to “Sample preparation of control group: sample 1 (ZSP is added to deionized water but not heavy metal ions) and sample 2 (only original metal solution is added).". Thank you again for your positive comments and valuable suggestions to improve the quality of our manuscript. We are looking forward to your positive decision.
Q2.As a suggestion, between points 2.4.1 and 2.4.3 the authors explain the dynamics of the tests to determine the influence of biomass concentration, temperature and pH; it is a sequential protocol that should be improved in the future by designing experiments. At the same time, section 2.4.3 does not specify the concentration of biomass used in the tests, nor the volume of metal solution. In addition, these three sections must specify the ultimate purpose of the experiments described: obtain the optimal operating conditions?.
Response:Thank you very much for your suggestion. The purpose of the experiment in 2.4.1, 2.4.2 and 2.4.3 of the manuscript has been re described and marked in blue. For the research on the interaction of biomass concentration, temperature, pH value and other factors on ZSP adsorption capacity, we agree that it is a very good research direction. However, due to the tight experimental funds and heavy graduation tasks, we have to present these experimental results to you temporarily. We will supplement these experimental results by redesigning experiments in the future. Thank you again for your valuable suggestions, which will be our focus in the future.
Q3. In section 2.4.4 the authors describe the procedure to obtain the kinetics, but do not detail the operating conditions. I imagine that it was the optimal conditions obtained in the previous tests, but this aspect must be perfectly explained. I understand that these “optimal conditions” are results and cannot be described in this section, but this important aspect must be clearly specified.
Response:Very sorry for our incorrect statement. We have added this description to section 2.4.4 of the manuscript and marked it in blue. We have corrected this erroneous statement.
Q4. The foregoing is applicable to point 2.4.5.
Response:Thanks for your advice. We have corrected this erroneous statement in section 2.4.5.
Q5. Sections 2.5.1 and 2.5.2 should be joined because the EDX sensor is attached to the SEM. In addition, the sample preparation protocol must be included before metallization with Au, that is, fixation and desiccation steps, if any. Any author who wants to reproduce the procedure must be clear about the steps to follow..
Response:Very sorry for our insufficient statement. We have merged 2.5.1 and 2.5.2 in the manuscript as a whole. In addition, we also added the steps of sample drying and fixation. And we hope our responds would satisfy you and thanks again for your kindly comments.
Q6: Section 2.5.3, something to say about sample preparation?
Response:Thank you for your thoughtful comments. We have rewritten the sample preparation process. The phrase “ZSP samples (1 mg) were ground with KBr powder (100 mg) and pressed into pellets for FT-IR measurement.” was changed to “Accurately weigh 1.00 mg of samples respectively, and grind them with dry KBr powder (100 mg) in a grinder until uniform powder were formed; Then, the powders were placed in the mold and pressed into transparent sheets with a tablet press.".
Q7. Section 2.6, specify the volume of cell suspension and ZPS suspension added to the wells. Also the volume of metallic solutions. Although this can be inferred, please specify how the controls were designed.
Response:Thank you very much for your suggestion. The volumes of cell suspension, ZPS solution and metal solution have been described and marked in blue in section 2.6 of the manuscript.
Q8. Section 3.2.3, the operating conditions do not appear (see comment no. 3).
Response:Thank you for your thoughtful comments. The relevant operating conditions have been described and marked in blue. The phrase “The result of contact time on adsorption behavior is shown in Fig. 3D.” was changed to “Based on the optimal adsorption conditions (biomass addition is 0.8 g/L; reaction temperature is 25 ℃; reaction pH is 4.0), we investigated the influence of contact time on the adsorption behavior, and the results are shown in Fig. 3D.” .
Q9. In section 3.2.4 it is important not to confuse “optimal biosorption capacity” with “maximum biosorption capacity”, which is a theoretical value produced by the Langmuir model. Please correct the error. In addition to the above, what was said in point 3 is also valid for this section since we do not know what operating conditions were used to obtain the isotherms, apart from the metal concentrations.
Response:Thank you for your thoughtful comments. We have corrected the error in section 3.2.4. At the same time, the operating conditions for obtaining isotherms have been added to this section. The above contents are marked in blue.
Q10.Sections 3.3.1 and 3.3.2 should be merged into a single section, titled for example “SEM-EDX Analysis”.
Response:Thank you very much for your suggestion. Sections 3.3.1 and 3.3.2 of the manuscript have been merged.
Q11.Section 3.4, the results should be better discussed in the real context of the findings. It is not possible to state that ZSP can inhibit cell death induced by heavy metal ions, in fact ZPS is capable of adsorbing heavy metals thus reducing their concentration in the medium and secondarily this fact may mean a reduction in the risk of cell damage. This comment is also applicable to the Conclusions section. On the other hand, supporting the comments with a bibliography can improve the discussion of section 3.4.
Response:Thank you very much for your suggestion. We have rewritten Section 3.4. The rewritten contents are as follows: It can be seen from Fig. 8 that the survival rate of HepG2 cells in ZSP preprotected group increased significantly. However, in the treatment group with heavy metal ions added without protection, the cell survival rate was significantly reduced. Li et al. used oyster ferritin (GF1) to pre-protect HFF-1 cells and MC3T3-E1 cells, and found that GF1 had a strong protective effect against heavy metal invasion due to its strong ability to enrich heavy metal ions [18]. It is no coincidence that ZSP polysaccharide has been shown to have adsorption ability for heavy metals and has no toxic effect on cells. Therefore, when heavy metal solutions were added to the cell culture medium, the pre-protective effect was exerted by the adsorption of heavy metals by the pre-added ZSP, which reduced the concentration of heavy metals in the medium or produced substances that could not cause cellular toxicity. This fact may imply a reduced risk of cellular damage. Meanwhile, it also proved that ZSP has a relatively strong protective effect against the invasion of heavy metal ions.
Q12.Unify the criteria established in the introduction with the final conclusions. In the introduction, the use of ZPS for the treatment of water contaminated by heavy metals destined for the food industry and agriculture is identified as the objective of the work. The objectives and conclusions must be related, building a coherent argument from the experimental findings.
Response:Thank you for your thoughtful comments. We have rewritten the objectives and conclusions of the introduction, so that the two parts can be better related. These contents have been marked in blue in the introduction and conclusion. Thanks again for your valuable comments. With your suggestions, the quality of our manuscript has been greatly improved. And we hope our responds would satisfy you and thanks again for your kindly comments.
Q13.Unify the font size in the figure captions.
Response: Thank you for your advice. We have corrected this..

Reviewer 2 Report
In the present manuscript, authors have reported the biosorption characteristics of a polysaccharide (ZSP) for removal of Pb2+, Cu2+ and Cd2+ from aqueous solution. The subject is interesting however there is need to improve the English of manuscript and also there are certain quarries which must be satisfied before final publication. My comments/queries are given in the reviewed manuscript attached.

Author Response
Responses to reviewers, comments
Thank you very much for your affirmation of our work and kindly comments. We have revised the manuscript carefully. The responses and modifications are as follows according to your comments. All revisions were marked in red in the revised manuscript and Supplementary Materials.
Since only one reply document can be uploaded, we combined one reply letter and one revised manuscript into a PDF file and then uploaded it. We are deeply sorry for the inconvenience caused to editors and reviewers.

Round 2
Reviewer 2 Report
Authors have incorporated all the suggested changes in the revised manuscript. The manuscript is recommended for publication after following changes.
1) Section 2.4 lines 4-6: Either rewrite properly or add the suggested lines as: “For preparation of control only ZSP was added to deionised water for sample 1 and in sample 2 only respective metal ion solutions were used. Supernatant was obtained after centrifugation (4000 r/min, 10 min) of sample solution.”
2) Section 2.4 lines 7, “was passed through a 0.22 m M…..” M is not defined here. If it is for meter then it must be “m”.
3) Font size of all equations should be aligned with text.
4) Section 2.4.1: Remove “initial” from heading
5) Section 2.4.1: The text is not clear. Either rewrite it or add the suggested text as: “The effect of dose of ZSP on metal ion biosorption was studied in order to obtain the optimal concentration of adsorbent. The respective metal ion solutions (10 mL each; 50 mg/L) were mixed with ZSP polysaccharide solutions of different concentrations (0.2 - 1.2 g/L) at room temperature.”
6) Sections 2.4.2 line 1: 10 mL of metal ion solution…..
7) Section 2.4.3 line 1: The pH of metal ion solution was…..
6) Section 2.4.5 line 1: The effect metal ion concentration…..
7) Section 2.4.5 lines 2-3: Text without brackets is suggested as: “Optimum conditions, as described in section 2.4.1 to 2.4.4, of the biomass addition, temperature, pH and time were used for isothermal adsorption study.”
8) Section 2.5.3 lines 1-3 should be written in past tense. Suggested text is: “Accurately weighed 1.00 mg of each sample was ground with dry KBr powder (100 mg) in a grinder until a uniform powder was obtained. Powder of respective sample was placed in the mold and pressed into transparent sheets with a tablet press.”
9) Section 2.6 Line 9: The ZSP solutions (10 µ L/hole) were…
10) Section 2.6 Line 11-12: “Concentrations of 20 mM of the three heavy metal ions
solution (10 µ L/hole) were added” Rewrite this text to make it clear.
11) Section 3.2.3 Lines 3-4: In figure 3D the maximum adsorption is at around 40 minutes not 30 minutes. “It was observed that the maximum removal of Pb2+, Cu2+ and Cd2+ ions on ZSP occurred in about 40 minutes.”
12) Section 3.2.4 Lines 2-3: Rewrite without brackets.
13) Section 3.2.4 Lines 5-8: It is not the removal rate of ZSP as written in text. Suggested text is: “Meanwhile, it was found that the removal rates of metal ions by ZSP decreased or tend to balance with an increase in metal ion concentrations after 75 mg/L, 100 mg/L and 100 mg/L for Pb2+, Cu2+and Cd2+, respectively.”
Author Response
Thank you again for your affirmation and kind comments on our work. With your professional, patient and meticulous suggestions, the quality of our manuscript has been greatly improved. According to your valuable comments, we carefully revised the manuscript again. The reply and modifications are as follows.
All revisions are marked in red in the revised manuscript and supplementary materials.
Responds to comments from Reviewer 2 (Round 2)
Q1. Section 2.4 lines 4-6: Either rewrite properly or add the suggested lines as: “For preparation of control only ZSP was added to deionised water for sample 1 and in sample 2 only respective metal ion solutions were used. Supernatant was obtained after centrifugation (4000 r/min, 10 min) of sample solution.”.
Response: Thank you for pointing out this. The phrase “Sample preparation of control group: sample 1 (ZSP is added to deionized water but not heavy metal ions) and sample 2 (only original metal solution is added). Take supernatant after centrifugation (4000 r/min,10 min) of sample solution.” was changed to “For preparation of control only ZSP was added to deionised water for sample 1 and in sample 2 only respective metal ion solutions were used. Supernatant was obtained after centrifugation (4000 r/min, 10 min) of sample solution.". Thank you again for your positive comments and valuable suggestions to improve the quality of our manuscript. We are looking forward to your positive decision.
Q2.Section 2.4 lines 7, “was passed through a 0.22 m M…..” M is not defined here. If it is for meter then it must be “m”..
Response:Thanks for pointing out this problem. We have changed "0.22 μM" to "0.22 μm".
Q3. Font size of all equations should be aligned with text.
Response:Done as requested.
Q4. Section 2.4.1: Remove “initial” from heading.
Response:Done as requested.
Q5. Section 2.4.1: The text is not clear. Either rewrite it or add the suggested text as: “The effect of dose of ZSP on metal ion biosorption was studied in order to obtain the optimal concentration of adsorbent. The respective metal ion solutions (10 mL each; 50 mg/L) were mixed with ZSP polysaccharide solutions of different concentrations (0.2 - 1.2 g/L) at room temperature.”.
Response:Thank you very much for your suggestion. We've rewritten it.
Q6: Sections 2.4.2 line 1: 10 mL of metal ion solution.
Response:Thank you very much for your thoughtful comments. We've rewritten it..
Q7. Section 2.4.3 line 1: The pH of metal ion solution was.
Response:Thanks for pointing out this problem. This problem has been corrected by us.
Q8. Section 2.4.5 line 1: The effect metal ion concentration.
Response:Done as requested.
Q9. Section 2.4.5 lines 2-3: Text without brackets is suggested as: “Optimum conditions, as described in section 2.4.1 to 2.4.4, of the biomass addition, temperature, pH and time were used for isothermal adsorption study.”.
Response:Thank you for your thoughtful comments. We have corrected the unclear statement in section 2.4.5.
Q10.Section 2.5.3 lines 1-3 should be written in past tense. Suggested text is: “Accurately weighed 1.00 mg of each sample was ground with dry KBr powder (100 mg) in a grinder until a uniform powder was obtained. Powder of respective sample was placed in the mold and pressed into transparent sheets with a tablet press.”.
Response:Thank you very much for your suggestion. We've rewritten it.
Q11.Section 2.6 Line 9: The ZSP solutions (10 µ L/hole) were.
Response:Done as requested.
Q12.Section 2.6 Line 11-12: “Concentrations of 20 mM of the three heavy metal ions
solution (10 µ L/hole) were added” Rewrite this text to make it clear..
Response:Thank you for your thoughtful comments. The phrase “Concentrations of 20 mM of the three heavy metal ions solution (10 μL/hole) were added to the cell culture plates” was changed to “The metal ion solutions (10 μ L each; 20mM) were added to the sample holes of cell culture plate respectively.”.
Q13.Section 3.2.3 Lines 3-4: In figure 3D the maximum adsorption is at around 40 minutes not 30 minutes. “It was observed that the maximum removal of Pb2+, Cu2+ and Cd2+ ions on ZSP occurred in about 40 minutes.”.
Response: Thank you for your advice. We have changed “about 30 minutes later” to “in about 40 minutes”.
Q14.Section 3.2.4 Lines 2-3: Rewrite without brackets..
Response: Thank you for your advice. We've rewritten it.
Q15.Section 3.2.4 Lines 5-8: It is not the removal rate of ZSP as written in text. Suggested text is: “Meanwhile, it was found that the removal rates of metal ions by ZSP decreased or tend to balance with an increase in metal ion concentrations after 75 mg/L, 100 mg/L and 100 mg/L for Pb2+, Cu2+and Cd2+, respectively.”.
Response: Thank you for your advice. We have corrected this.
